# Adenosine Monophosphate-Activated Protein Kinase (AMPK) Phosphorylation Is Required for 20-Hydroxyecdysone Regulates Ecdysis in *Apolygus lucorum*

**DOI:** 10.3390/ijms24108587

**Published:** 2023-05-11

**Authors:** Yongan Tan, Liubin Xiao, Jing Zhao, Jieyu Zhang, Sheraz Ahmad, Dejin Xu, Guangchun Xu, Linquan Ge

**Affiliations:** 1Institute of Plant Protection, Jiangsu Academy of Agricultural Sciences, Nanjing 210014, China; xlb@jaas.ac.cn (L.X.); jingzhao0126@126.com (J.Z.); 18362825416@163.com (J.Z.); jaasxdj@jaas.ac.cn (D.X.); jaasxgc@jaas.ac.cn (G.X.); 2College of Plant Protection, Yangzhou University, Yangzhou 225009, China; dh20010@yzu.edu.cn

**Keywords:** *Apolygus lucorum*, AMPK, 20E, phosphorylation, ecdysis

## Abstract

The plant mirid bug *Apolygus lucorum* is an omnivorous pest that can cause considerable economic damage. The steroid hormone 20-hydroxyecdysone (20E) is mainly responsible for molting and metamorphosis. The adenosine monophosphate-activated protein kinase (AMPK) is an intracellular energy sensor regulated by 20E, and its activity is regulated allosterically through phosphorylation. It is unknown whether the 20E-regulated insect’s molting and gene expression depends on the AMPK phosphorylation. Herein, we cloned the full-length cDNA of the *AlAMPK* gene in *A. lucorum*. *AlAMPK* mRNA was detected at all developmental stages, whereas the dominant expression was in the midgut and, to a lesser extent, in the epidermis and fat body. Treatment with 20E and AMPK activator 5-aminoimidazole-4-carboxamide-1-β-d-ribofuranoside (AlCAR) or only AlCAR resulted in activation of AlAMPK phosphorylation levels in the fat body, probed with an antibody directed against AMPK phosphorylated at Thr172, enhancing *AlAMPK* expression, whereas no phosphorylation occurred with compound C. Compared to compound C, 20E and/or AlCAR increased the molting rate, the fifth instar nymphal weight and shortened the development time of *A. lucorum* in vitro by inducing the expression of *EcR-A*, *EcR-B*, *USP*, and *E75-A*. Similarly, the knockdown of *AlAMPK* by RNAi reduced the molting rate of nymphs, the weight of fifth-instar nymphs and blocked the developmental time and the expression of 20E-related genes. Moreover, as observed by TEM, the thickness of the epidermis of the mirid was significantly increased in 20E and/or AlCAR treatments, molting spaces began to form between the cuticle and epidermal cells, and the molting progress of the mirid was significantly improved. These composite data indicated that *AlAMPK*, as a phosphorylated form in the 20E pathway, plays an important role in hormonal signaling and, in short, regulating insect molting and metamorphosis by switching its phosphorylation status.

## 1. Introduction

*Apolygus lucorum*, a global Miridae pest, occurs throughout Asia, Europe, Africa, and America, and causes significant economic losses yearly [1]. The resurgence of *A. lucorum* calls for more research to mitigate its damaging effects without compromising Bacillus thuringiensis cotton cultivation [2,3], and chemical control strategies involving various insecticides remain the preferred option for controlling *A. lucorum*, but the extensive application of insecticides has resulted in the development of *A. lucorum* resistant to these chemicals and is potentially harmful to the environment [2,3]. 

Sterol hormone 20-hydroxyecdysone (20E) is specifically biosynthesized from cholesterol and produced through the activation of AMPK and PI3K pathway, under the catalysis of a series of cytochrome P450 enzymes and is pivotal for insect molting and metamorphosis [4,5]. Ecdysone, the precursor of 20E, is synthesized and secreted by a pair of prothoracic glands in holometabolous insects and then released into the hemolymph, where it is converted to the active form 20E in peripheral tissues such as fat body, midgut, and malpighian tubules during larval stages [6,7]. The presence of ecdysone in the ovary was first identified more than 40 years ago, and now it is well-known that ovarian follicular cells produce ecdysone from scratch [8]. However, in some Lepidoptera, male gonads also release moderate amounts of ecdysone in vitro [9]. Furthermore, by binding the receptor complex of *EcR-USP*, 20E rapidly and highly induces the expression of primary-response genes, including nuclear receptors or transcription factors such as *E75*, *E93*, *Br-C*, and *E74*. Finally, the nuclear receptor complex of *EcR-USP* triggers a transcriptional cascade that induces molting and metamorphosis in insects [10,11,12]. 20E massively induces the expression of three different *BmE75* isoforms in *Bombyx mori*, which consequentially coordinates feedback to 20E biosynthesis [10]. *E93*, encoded by a member of the helix-turn-helix (HTH) transcription factor family, is involved in the crosstalk of 20E signaling with juvenile hormone (JH) signaling via the JH primary response gene Kr-h1 [11]. Notably, 20E signaling predominantly mediates the occurrence of autophagy in larval tissue and organs during larval molting and larval-pupal metamorphosis [13]. 

Adenosine monophosphate-activated protein kinase (AMPK) is a heterotrimeric serine/threonine kinase complex comprised of one catalytic subunit α and two regulatory subunits βγ [14]. AMPK is mainly activated in response to physiological elevation of the AMP/ATP or ADP/ ATP ratio caused by energy deprivation and low nutrient levels and activates 20E in insects [15]. Furthermore, the kinase activity of AMPK is induced when the upstream kinases liver kinase B1 (LKB1) and calmodulin-dependent protein kinase-kinase β (CaMKKβ) phosphorylate AMPK on a conserved threonine residue within the activation loop of α subunit [16]. Under low intracellular ATP levels, AMP binds to the γ subunit of AMPK and further leads to a conformational change that promotes phosphorylation and inhibits dephosphorylation of Thr^172^ [17]. As a conservative eukaryotic energy sensor to restore intracellular ATP homeostasis, AMPK activation reprograms metabolism by switching off anabolic signaling pathways while turning on catabolic signaling pathways [18].

AMPK is highly conserved throughout eukaryotes, and its activity is regulated allosterically by AMP and through phosphorylation at Thr^172^ [19]. However, it is still unknown whether the 20E-regulated insect molting and gene expression depend on AMPK phosphorylation [20]. In the present study, we cloned the homolog of the *A. lucorum AlAMPK* gene and characterized its developmental and spatial expression profile using qRT-PCR and western blot hybridization. This study shows that 20E-induced AlAMPK phosphorylation and AlAMPK phosphorylation is essential for ecdysis and metamorphosis in *A. lucorum*.

## 2. Results

### 2.1. Characteristics of AlAMPK Cloned from A. lucorum

Based on the SMART cDNA library for *A. lucorum* and RACE, two fragments corresponding to the 5′ and 3′ ends of *AlAMPK* cDNA were detected. A 1980-bp nucleotide sequence representing the complete *AlAMPK* cDNA sequence was deposited in the GenBank database (MN514867). The full-length *AlAMPK* cDNA includes a 287-bp 5′-untranslated region (UTR), a 139-bp 3′-UTR, a canonical polyadenylation signal sequence (CGATAA), a poly (A) tail, and a 1554-bp ORF. The ORF encodes a polypeptide comprising 517 amino acids, with a predicted molecular weight of 58.78 kDa and a theoretical isoelectric point of 7.10. A search of the National Center for Biotechnology Information (NCBI) Conserved Domain Database (https://www.ncbi.nlm.nih.gov/cdd) (accessed on 30 March 2023) revealed three conserved domains, namely the S-TKc, UBA-AID-AMPKalpha domain, and AMPKA-C domains. The S-TKc domain (amino acids 18-270) is the catalytic domain of serine/threonine protein kinase. UBA-AID-AMPK α domain (amino acids 287-351aa), which is a ubiquitin-associated (UBA)-like autoinhibitory domain (AID) found in vertebrate 5’AMP-activated protein kinase catalytic α (AMPKα) subunits. AMPKA-C domain (amino acids 402-515), which is the C-terminal regulatory domain of 5’AMP-activated protein kinase (AMPK) α catalytic subunit, mainly involved in the formation of AMPK heterotrimers (Figure 1A). Sodium dodecyl sulfate polyacrylamide gel electrophoresis (SDS-PAGE) and fluorography demonstrated that the molecular weight of the in vitro protein was similar to the expected 60 kDa (Figure 1B). A phylogenetic analysis based on the *AlAMPK* sequence uncovered 20 *AMPK* genes in the NCBI database (Figure 2). Comparison of predicted N-terminal amino acid sequences of the phosphorylation domains of *A. lucorum* AMPK with their counterparts from Drosophila, mouse, and human are presented in Appendix A.

### 2.2. AlAMPK Was Phosphorylated by 20E

The effect of 20E on AlAMPK phosphorylation was studied in detail by examining the level of AMPK phosphorylation at Thr^172^. Phosphorylation that occurred at Thr^172^ was examined in *A. lucorum* AMPK through Western blot. Auto-phosphorylation of AlAMPK was significantly enhanced in the fat bodies that were cultured in vitro with 20E as compared to the control groups (Figure 3). Further evidence that the phosphorylated protein is AMPK and that the antibody specifically recognizes AMPK comes from the use of the AMPK inhibitor, compound C, and the AMPK activator, AlCAR. In vitro treatment of fat bodies from the second instar nymph with compound C for 24 h revealed a substantial reduction in antigen recognition by the anti-phospho-AMPK antibody. In contrast, treatment with 20E and AlCAR greatly increased immune activity. The total protein level was also checked using an anti-AlAMPK antibody, and results showed that it did not change with stimulation by 20E. This result indicated that the 20E could induce the phosphorylation of AlAMPK.

### 2.3. AlAMPK Expression Profile in the Ecdysis Stage

By quantitative real-time PCR, the expression of *AlAMPK* was measured in all developmental stages, including the first day of nymph to the second day of adulthood. The results showed that *AlAMPK* was continuously expressed throughout the whole life cycle of *A. lucorum*, with peaks correlating with the ecdysteroids pulse (Figure 4A). The spatial expression pattern of *AlAMPK* mRNA was analyzed by RT-PCR using total RNA prepared from five tissues in third-instar nymphs. The transcript levels of *AlAMPK* were higher in the midgut, epidermis, and fat body; in contrast, only small amounts of mRNAs for AlAMPK were detected in flying muscle and Malpighian tubules (Figure 4B).

### 2.4. AlAMPK Expression and Phenotypes under Spraying and RNAi Experiments

We investigated the functions of *AlAMPK* in *A. lucorum* development time and molting rate when knockdown of the mRNA expression of *AlAMPK* by spraying and RNAi experiments, respectively. Activation of AMPK phosphorylation by 20E and/or AlCAR in vitro enhanced the expression of *AlAMPK* in the nymph of *A. lucorum* compared to the compound C treatment (Figure 5A). Similarly, in the fat body of surviving *A. lucorum* nymphs after 48 h injection, qRT-PCR analysis confirmed that *AlAMPK* expression level was significantly reduced by treatment with ds*AlAMPK* compared to treatment with dsGFP and water control (Figure 5B). In the 20E spray treatment, the weight of the fifth instar nymph was 4.79 ± 0.17 mg, and the weights decreased by 10.21% and 4.23% after treatments with compound C and water, respectively. In addition, the weights decreased by 7.64% in ds*AlAMPK* compared to 20E injections (Figure 5C,D). In the spraying treatment, the AlCAR showed a high molting rate of up to 89.1%, followed by the 20E, 20E + AlCAR combined treatment, whereas the notable lowest role was observed by AlCAR + compound C combined treatment up to 40.3%. Similarly, in response to injection treatment, the 20E was found to play a highly influential role, reaching 89.6%, followed by ds*GFP* at 86.6%, whereas no influential role of compound C and ds*AlAMPK* was observed (Table 1).

### 2.5. The 20E-Induced Gene Expression under the Spray of Seven Different Compounds

Herein, we also investigated the effect of seven treatments inducing the transcription of *AlAMPK* in *A. lucorum* (Figure 6A). Among these seven different compounds, the highest expression of *ECR-A* was recorded under 20E + AlCAR where it reached a maximum of 3.5 folds. Induced expression of *ECR-A* was also observed under 20E and AlCAR compared to water. In contrast, the transcription level plummeted significantly following the 20E + Compound C, AlCAR + Compound C, and Compound C treatment. A similar expression trend was noticed for *ECR-B*, *USP*, and *E75-A* (Figure 6B,C).

### 2.6. The 20E-Induced Gene Expression under the AMPK RNAi

The ds*AlAMPK* significantly affected the expression level of four 20E-related genes. For instance, the expression of *ECR-A* induced under 20E injection, in contrast, was inhibited in ds*AlAMPK* treatment. Similarly, the *ECR-B* displayed reduced mRNA levels in ds*AlAMPK* compared to the 20E injection. The *USP* and *E75-A* gene plummeted substantially in ds*AlAMPK*, confirming the association of the *AMPK* gene in 20E-mediated ecdysis in *A. lucorum* (Figure 7). 

### 2.7. Observation of Epidermal Structure under Spray and RNAi by TEM 

We also surveyed the variation of the epidermal structure in the 3rd instar nymphs of *A. lucorum* after spraying with seven chemical compounds or injection with dsRNA by TEM. The results indicated that the thickness of the nymph epidermis in 20E and/or AlCAR treatment was significantly increased, the ecdysial space started to form between the cuticle and epidermal cells, and molting progress of nymphs was improved considerably in *A. lucorum,* which compared to the compound C treatment (Figure 8). Similarly, the cuticle was closely connected with the epidermic cells, the ecdysial space has not yet formed after knockdown of Al*AMPK* by RNAi (Figure 9).

## 3. Discussion 

The *AMPK* gene is a member of the gamma subunit family and is found in all domains of life. In eukaryotic cells, AMPK is a key regulatory enzyme of cellular energy homeostasis and is involved in regulating a diverse range of metabolic pathways [21]. AMPK is a serine/threonine kinase and is highly conserved throughout eukaryotes, regulated allosterically by AMP and through phosphorylation at Thr^172^ [22]. AMPK, in addition, also plays an important role in insect development, tissue growth, molting, and autophagy [23]. In recent decades the AMPK gene has been functionally characterized in numerous organisms such as *Hyphantria cunea* [24], *B. mori* [25], *Drosophila melanogaster* [26], *Homo sapiens* [27], and so on; however, the 20E-induced AMPK phosphorylation drives developmental programming, ecdysis, and metamorphosis in *A. lucorum* have not been properly understood.

### 3.1. AMPK Fine-Tune A. lucorum Physiology

The key role of AMPK genes in the regulation of various key parameters has been previously reported. For instance, the ecdysteroids signaling plays a major role in insect molting and metamorphosis, which is triggered by the binding of 20E to a heterodimer composed of EcR and ultra spiracle (USP) [24]. The present study revealed the AMPK role in inducing the physiological parameters of *A. lucorum* by dominant expression in all developmental stages with peaks correlating with the ecdysteroids pulse. Additionally, the silencing of AlAMPK by spray and RNAi delayed the nymphal growth and weight, contrary to the activation of AMPK phosphorylation by 20E and/or AlCAR, which enhanced the expression of *AlAMPK* in the nymph of *A. lucorum*. It is well known that 20E acts through insect larvae’s central nervous system (CNS) to induce wandering behavior and escape from food [28,29]. In addition, 20E subtly alters the feeding behavior of insects and, consequently, their food intake [30]. However, the induction of wandering behavior and the reduction of feeding behavior can lead to energy stress, such as sugar starvation. This, in turn, ultimately leads to an increase in the cellular AMP/ATP ratio, which activates AMPK and induces developmental parameters [31,32,33]. Our findings suggest a potential role of *AlAMPK* in *A. lucorum* development time and molting rate. The *AlAMPK* was expressed in all developmental stages and in all tested tissues, whereas the dominant expression was observed in the epidermis and midgut. Furthermore, when the mRNA was silenced with ds*AlAMPK* injection treatment, the size of the first instar nymph and second instar nymphs were reduced. Additionally, the qRT-PCR analysis confirmed that treatment with ds*AlAMPK* reduced *AlAMPK* expression in the fat body of surviving *A. lucorum* nymphs, compared to ds*GFP* and untreated treatments. Besides, 20E also increases *AlAMPK* mRNA expression level. A previous study showed that 20E does not directly induce fat body lipolysis in *B. mori* [34]. 20E reduces food consumption via an unidentified tissue (such as the brain or midgut), causing starvation and fat body lipolysis during molting and pupation in *B. mori* [29]; compared to the previous finding, the results of this study, we confirmed that treatment with ds*AlAMPK* reduced *AlAMPK* expression in the fat body of surviving *A. lucorum* nymphs.

### 3.2. 20E-Induced AMPK Phosphorylation Is Essential for Molting in A. lucorum

In insects, the regulation of development, molting, and metamorphosis is coordinated by various endocrine hormones and cellular signals, 20E, and the most prominent AMPK phosphorylation [35]. For instance, mitochondrial health is critical for skeletal muscle function and is improved by exercise training through mitochondrial biogenesis and removing damaged/dysfunctional mitochondria via mitophagy [36]. These changes were monitored using a novel fluorescent reporter gene, pMitoTimer, that allowed assessment of mitochondrial oxidative stress and mitophagy in vivo and were preceded by increased phosphorylation of AMPK at tyrosine 172 and of unc-51 like autophagy activating kinase 1 (Ulk1) at serine 555. Using mice expressing dominant negative and constitutively active AMPK in skeletal muscle, we demonstrate that Ulk1 activation depends on AMPK [37]. Furthermore, the study of Zhao et al. (2023) found that AMPK phosphorylation activated the BmAtg1c mRNA expression and BmAtg1c protein expression, enhancing the autophagy simultaneously peaked in the fat body during larval-pupal metamorphosis [38]. Additionally, 20E activates AMPK in the insect fat body in two ways: by up-regulating the mRNA levels of all three *AMPK* subunits and inducing energy stress to activate AMPK [39]. The transcription levels of all three AMPK subunits, the protein level of AMPK, and the phosphorylation level of AMPK were all elevated in the *B. mori* fat body and the *D*. *melanogaster* fat body during pupariation, consistent with 20E signaling. Gain-of-function and loss-of-function experiments showed that 20E activates AMPK transcriptionally [39]. In line with several other studies have been conducted but the potential role of AMPK phosphorylation in the molting of *A. lucorum.* Our study, for the first time, revealed the essential role of AMPK phosphorylation that induces molting also the activation of AMPK phosphorylation by 20E and/or AlCAR in vitro*,* enhanced the expression of *AlAMPK* in the nymph of *A. lucorum*, which compared to the compound C treatment. Additionally, also in the fat body of surviving *A. lucorum* nymphs after 48 h post-injection the qRT-PCR analysis confirmed that *AlAMPK* expression level was significantly reduced by treatment with ds*AlAMPK* in contrast to ds*GFP* and untreated. Quantitative real-time PCR measured *AlAMPK* expression from nymph to adult. *AlAMPK* expression peaks correspond to ecdysteroids pulses in the *A. lucorum* life cycle. RT-PCR analyzed five tissues for *AlAMPK* mRNA in the midgut, epidermis, and fat body transcript levels were higher than flying muscle and malpighian tubules in third-instar nymph. Although TEM revealed the variation of the epidermal structure in the surveyed third instar nymphs of *A. lucorum* after seven chemical compounds spraying or dsRNA injection. Furthermore, the thickness of the nymph epidermis in 20E and/or AlCAR treatments was significantly increased, the ecdysial space started between the cuticle and epidermal cells, and the molting progress of nymphs was improved considerably in *A. lucorum* in contrast to compound C treatment (Figure 8). Furthermore, previous studies suggested that AMPK activates when AMP levels increase in the body with decreased ATP levels and activated AMPK inhibits anabolic processes and promotes catabolism to minimize ATP utilization while promoting ATP production [39,40]. In response, our study found that the third instar nymphs’ integument ultrastructure after drip administration showed cuticle and epidermic cell effects. The apical plasma membrane area of nymph epidermic cells became smooth, the microvilli on epidermal cells’ apical plasma membrane reduced, and the ecdysial space formed between the cuticle and epidermal cells. Additionally, a dense nymphal cuticle and microvilli were observed after 20E + AlCAR and 20E treatment, also a thickened epidermis separated from epidermal cells and enlarged the ecdysial space, allowing ecdysial droplets to form a new cuticulin layer on the contrary alone 20E and AlCAR treatment resulted in narrow ecdysial.

Herein, we report that 20E is essential for ecdysis and development, and the TEM observation of epidermal structure of *A. lucorum* confirmed that the epidermic cells of the nymph in 20E-treated, untreated and ds*GFP*-treated were separated from the epidermis, the ecdysial space started to form between the cuticle and epidermal cells, and the ecdysial droplets were free in the ecdysial space. In 20E-treated, the apical plasma membrane of epidermic cells formed many bulges and a new cuticulin layer. There were sparse microvilli in the apical plasma membrane of epidermic cells in untreated and ds*GFP*-treated. In ds*AMPK*-treated, the cuticle was closely connected with the epidermic cells, the microvilli remained developed and dense. Overall, the molting progress of nymphs in the 20E-treated was significantly faster than that in the other treatment groups, and the molting progress in ds*AMPK*-treated was the slowest; collectively, these results suggested that the 20E is essential for ecdysis and development in *A. lucorum.* Taken together, our findings indicate that pathways activated in parallel by this agent then concomitantly activate AlAMPK. These results indicate that actions ascribed to AMPK after AlCAR treatment may be influenced by the concomitant modulatory actions of 20E. Collectively, these results suggested that AMPK phosphorylation is essential for molting in *A. lucorum.*

### 3.3. The Induction/Inhibition of AMPK Phosphorylation Sensitizes the Expression of 20E Downstream Genes

The steroidal hormone 20E is primarily secreted in an insect’s brain through the prothoracicotropic hormone, which further stimulates the prothoracic gland to synthesize ecdysone and plays an essential role in growth and development [39,41]. The active metabolite of ecdysone and 20E works through ecdysone receptor (EcR) and USP, *E75-A* and *E75-B,* to initiate molting and metamorphosis by regulating downstream genes [34]. Previously, the study of Roy et al. (2012) found that EcR was expressed in the PTTH-producing neurosecretory cells (PTPCs) in the larval brain of *B. mori*, suggesting that PTPCs function as the master cells of development under the regulation of 20E [42]. The present study revealed the hindered expression of 20E downstream genes in which ECR-B was observed with the dominant expression under 20E, AlCAR, and 20E + AlCAR combined treatment. However, the lower expression was followed by the other three genes under the same treatment. Notably, the expression plummeted in the *ECR-A*, *ECR-B*, *USP*, and *E75-A* genes under different treatments after induction/inhibition of AMPK phosphorylation. Collectively, these results revealed that manipulation with *AMPK* genes directly alters the functions of 20E downstream genes; however, the underlying molecular mechanisms need to be further studied.

## 4. Materials and Methods

### 4.1. Experimental Insect Rearing

*A. lucorum* were collected from *Vicia faba* grown in fields in Yancheng (33.110 N, 120.250 E) (Jiangsu China) from July to August 2020 and reared on *Phaseolus vulgaris* with a 10% sucrose solution in an incubator at 25  ±  1 °C with 70  ±  5% humidity under a 14: 10-h light/dark cycle.

### 4.2. Cloning of AlAMPK Gene

The total RNA of ten third instar nymphs was isolated using the SV Total Isolation System kit (Promega Corporation, Madison, WI, USA). First-strand cDNA was synthesized using the PrimeScript™ 1st Strand cDNA Synthesis Kit (TaKaRa Biotechnology, Dalian, Co., Ltd. Dalian, China) according to the manufacturer’s instructions. The first-strand cDNA obtained from the insect was used as the PCR template. The gene-specific primers for *AlAMPK* were designed based on conserved regions found in AMPK from other insect species: *Cimex lectularius* (GenBank accession No. XM_014407037), *Halyomorpha halys* (XM_024358872) and *Myzus persicae* (XM_022324589). The primers were AMPK-F and AMPK-R (Appendix A). Following amplification, the products were separated on a 1.0% agarose gel and purified using the TIANgel Midi DNA Purification System (TianGen Biotech CO., LTD, Beijing, China). Purified DNA fragments were cloned into the pEASY-T3 cloning vector (TransGen Biotech, Beijing, China). Recombinant plasmids were isolated using the Plasmid Mini kit and sequenced. The full-length cDNA of *AlAMPK* was obtained by the rapid amplification of cDNA ends (RACE) using a SMART**^TM^**RACE cDNA Amplification Kit (Clontech, Palo Alto, CA, USA). PCR products were purified, cloned, and sequenced as above. All the primers used in this study are listed in (Appendix A).

### 4.3. In Vitro Translation

A pET28a vector containing the cDNA sequence encoding the *AlAMPK* gene was constructed and transformed into *Escherichia coli* BL21. The recombinant target protein was over-expressed and purified using nickel-nitrilotriacetic acid agarose according to the manufacturer’s protocol (ZoonBio Corporation, Nanjing, China).

### 4.4. Reagents, Proteins, and Antibodies

20E, permeable AMPK activator 5-aminoimidazole-4-carboxamide-1 β-d-ribofuranoside (AlCAR), and an AMPK inhibitor compound C were purchased from Sigma (Sigma-Aldrich LLC, Darmstadt, Germany). The reagents were dissolved in DMSO to obtain the desired concentrations and stored at −20 °C. Different reagents were used for 1 h before treatment was administered. The spraying assay consisted of seven experimental groups: (1) 20E (1.0 μmol/L) and AlCAR (25 μmol/L), (2) 20E only, (3) AlCAR only, (4) compound C (25 μmol/L) only, (5)20E and compound C, (6) AlCAR and compound C, and (7) water as a control.

Phospho-specific antibodies were developed and are commercially available for human AMPKα phosphorylated at Thr^172^ (cat. no. 2535, Cell Signaling Technology, Boston, MA, USA), as phosphorylation is essential for the activity of these enzymes. Comparison of deduced amino acid sequences of the phosphorylation domain of *A. lucorum* AMPK with their counterparts from Drosophila, humans, and mice showed identical phosphorylation sites, indicating high conservation among species (Appendix A). The consistency of phosphorylation sites among species suggests it is possible that commercial antibodies against mammalian kinase can be successfully used to investigate the phosphorylation of *A. lucorum* AMPK. Therefore, a commercial polyclonal antibody against mammalian AMPK phosphorylated at Thr^172^ was used in this present study. In addition, the antibodies against AlAMPK produced from the cDNA of *AlAMPK* ORF were preserved in our lab, used at 1:1000 dilutions, and incubated for 1 h at 37 °C.

### 4.5. Expression Profiles of AlAMPK mRNA Profile

qPCR was performed to profile the expression of *AlAMPK* across developmental stages (1st—5th instar nymph and female adult) and from selected tissues, including the epidermis, midgut, fat body, flying muscle, and Malpighian tubules. Total RNA was isolated using an SV Total Isolation System kit (Promega(Beijing) Biotech Co., Ltd., Beijing, China), and cDNA was synthesized as described above. mRNA levels were quantified by qPCR using the One Step SYBR PrimeScript RT-PCR Kit (Takara, Dalian, Liaoning, China) using the Bio-radi Cycler real-time quantitative RT-PCR detection system and the iCycleriQ real-time detection system software (version 3.0a; Bio-Rad Laboratories, Inc. Mississippi, USA) and hereafter amplification, the target gene cycle threshold (Ct) values were normalized to the reference gene by the 2^−ΔΔCT^ method (Livak and Schmittgen, 2001) [43]. Due to the low RNA quantity from individual nymphs or adults, a mixture of 15 whole bodies of *A. lucorum* at each developmental stage and 25 pooled tissues of each organ were used as one sample, respectively. Each experiment was replicated three times with independent sample groups. The *A. lucorum β-actin* (JN616391) was used as an endogenous reference gene for data normalization. Primers for amplifying a 131 bp *β*-*actin* and 125 bp *AlAMPK* were selected and used for qPCR (Appendix A). The mRNA expression profiles of *AlAMPK* across developmental stages and different tissues were assayed following the same procedure of Tan et al. (2014 a, b) [44,45].

### 4.6. Western Blot

Western blot was performed using proteins extracted from the fat body of surviving *A. lucorum* nymphs to identify phosphorylation of AlAMPK at the protein level after the above seven reagent treatments. The total proteins were extracted using a Tissue Protein Extraction Reagent Kit (Nanjing, ZoonBio Tech, Co., Ltd. Nanjing, China) according to the manufacturer’s instructions, and Bradford’s method was used to determine the protein concentration [46]. The Western blot analysis for each AlAMPK phosphorylation followed protocols described in previous studies [47].

### 4.7. dsRNA Synthesis and Application

Sense and antisense primers, including a T7 RNA Polymerase promoter, were designed based on the sequences of *AlAMPK* (Appendix A). PMD-19T Vector (Takara) plasmids harboring Ace gene segments served as templates for generating double-stranded RNAs (dsRNAs) targeting *AlAMPK*, utilizing the T7 RiboMAX Express RNAi System (Promega) as directed by the manufacturer. The concentration of dsRNA resuspended in DEPC-treated water was adjusted to 2.0 μg/μL, and dsRNA was kept at −70 °C for further experiments. *GFP* dsRNA (ds*GFP*) used as a control was obtained as described above. Freshly molted second instar nymphs were administered 2.5 μL of *AlAMPK* dsRNA by injection. Three independent experiments were carried out with 300 insects per group. Control animals were administered ds*GFP* and untreated. After dsRNA injection, the surviving 48 h old nymphs were dissected for fat body removal. The fat bodies were frozen immediately in liquid nitrogen and stored at −80 °C. RNA samples obtained from four individuals’ fat body tissues were tested, each in triplicate.

### 4.8. Detection of the Effect of Spraying and RNAi

After spraying and dsRNA injection, the mRNA expression in fat bodies of *AlAMPK* was analyzed by RT-PCR of *A. lucorum*, respectively. Nymphs were individually placed in 5 cm high, 1.5 cm diameter glass vials covered with a nylon screen. Each glass vial contained a green bean and a 1 × 5 cm wet paper strip for food and water. Nymphal development and molting rate were recorded daily until adult molting or death occurred. 80 to 100 nymphs were included per treatment and control. Nymphal weights of newly molted (<4 h old) fifth-instar nymphs were recorded. In each treatment and control, 20 nymphs were weighed individually. 

### 4.9. Transmission Electron Microscopy (TEM) after Spray and RNAi

About 24 h after molting, 110 third instar nymphs were collected into plastic cases with green beans. A topical application of 1 µL of each solution was applied on the pronotum of third-instar nymphs by microliter syringes (1 µL, Gaoge, Shanghai, China) (drip administration). Ten nymphs were treated in each group. The assay consisted of different experimental treatments described above in Section 2.4 and Section 2.7 The thoracic epidermis of nymphs treated with the chemical solution for 12 h was collected and fixed in 2.5% glutaraldehyde (pH 7.2, phosphate-buffered). The thoracic epidermis was fixed in 2.5% glutaraldehyde for 8 h, rinsed three times with 0.1 M phosphate buffer for 10 min each, and post-fixed in 1% osmium tetroxide for 2 h. Then the specimens were again rinsed three times with 0.1 M phosphate buffer for 10 min each, dehydrated separately in a series of acetone solutions (50%, 70%, 80%, and 90%) for 15 min each, and then 100% acetone (3 × 30 min). After that, the specimens were preserved in an embedding plate containing a pure embedding agent. The embedded plates were polymerized at 37 °C, 45 °C, and 60 °C for 48 h, respectively, and sectioned into 70–90 nm sections. Ultrathin sections (70 nm) were stained with uranyl acetate followed by lead citrate for 10 min and observed with a Hitachi H7650 transmission electron microscope.

### 4.10. Expression Profiles of Four 20E-Regulated Genes

The mRNA expression levels of *AlUSP* (JX675574), *AlE75A* (KX912697), *AlEcR-A* (KM401656), and *AlEcR-B* (KM504955) in fat body specimens from survived 48 h old nymphs *A. lucorum* after dsRNA injection and spraying treatments were assessed as reported by [47]. All the primers used are listed in (Appendix A).

### 4.11. Statistical Analysis

Differences in development time and nymphs’ weight were assessed by one-way ANOVA followed by Tukey’s HSD test (*p* < 0.05; Statistical Analysis System version 10). Duncan’s new multiple-range test assessed gene expression differences. Finally, GraphPad Prism software (version 9.4.1, Inc., LA Jolla, CA, USA) was used, following the same procedure as Ahmad et al. (2021) [47] for graphical representation following the procedure.

## 5. Conclusions

The obtained results revealed that the *AlAMPK* regulation is crucial for ecdysis and development in *A. lucorum*. The full-length cDNA sequence of the *AlAMPK* gene was cloned and functionally characterized. The *AlAMPK* showed expression in all developmental stages and all tested tissues. Additionally, the knockdown of *AlAMPK* by RNAi delayed the nymph molting rate, reduced nymphal weight and development time, and blocked the expression of 20E downstream genes. Additionally, the 20E and AlCAR induced the thickness of the nymphal epidermis, and the molting nymphs were significantly improved. The present study clearly shows that 20E-induced AlAMPK phosphorylation is essential for ecdysis and metamorphosis in *A. lucorum*. Further studies are required to fully understand the regulation and pathway mechanism of phosphorylation of 20E downstream genes that regulate the ecdysis in *A. lucorum.*

## Figures and Tables

**Figure 1 ijms-24-08587-f001:**
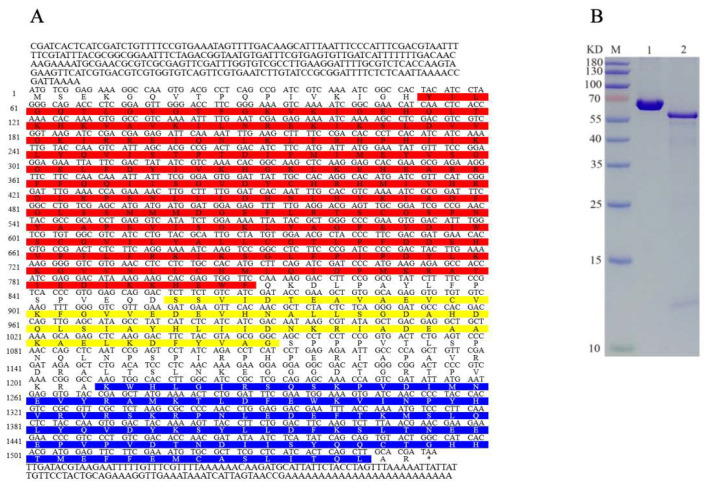
(**A**) *AlAMPK* cDNA and amino acid sequences. Bright red, yellow, and blue shaded amino acids indicate the S-TKc (18–270 aa), UBA-AID-AMPK alpha (287–351 aa), and AMPKA-C (402–515 aa), respectively. (**B**) Translation of AlAMPK in vitro. The AlAMPK translation product was resolved by 10% SDS-PAGE, followed by autoradiography. Lane 1: 0.5 mg/mL BSA, Lane 2: purified AlAMPK protein. Molecular masses of standards are shown on the left.

**Figure 2 ijms-24-08587-f002:**
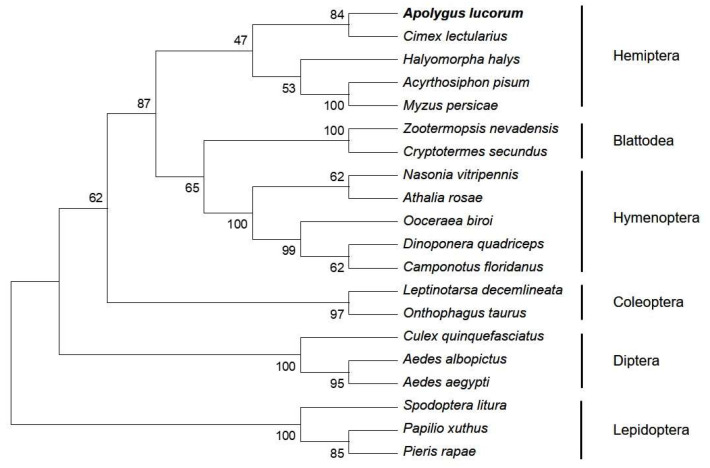
The phylogenetic analysis of AMPK between *A. lucorum* with other insects’ species. The neighbor-joining (NJ) method was employed for tree generations. The Jones–Taylor–Thornton (JTT) substitution model was applied with 1000 bootstrap replicates. The gamma law assessed between-site heterogeneity. The GenBank accessions of the proteins are as follows: *Apolygus lucorum* AlAMPK (MN514867), *Cimex lectularius* ClAMPK (XP_014262523), *Halyomorpha halys* HhAMPK (XP_024214640), *Acyrthosi phonpisum* ApAMPK (XP_008183803), *Myzus persicae* MpAMPK2 (XP_022180281), *Zootermopsis nevadensis* ZnAMPK (XP_021916442), *Cryptotermes secundus* CsAMPK (XP_023710112), *Nasonia vitripennis* NvAMPK (XP_001599874), *Athalia rosae* ArAMPK (XP_012268666), *Ooceraea biroi* ObAMPK (XP_011329194), *Dinoponera quadriceps* DqAMPK (XP_014480794), *Camponotus floridanus* CfAMPK (XP_011259007), *Leptinotarsa decemlineata* LdAMPK (XP_023020472), *Onthophagus taurus* OtAMPK (XP_022908510), *Culex quinquefasciatus* CqAMPK (EDS36926), *Aedes albopictus* AaAMPK (XP_029719110), *Aedes aegypti* AaAMPK (XP_001652572), *Spodoptera litura* SlAMPK (XP_022832070), *Papilio xuthus* PxAMPK (KPJ05221), *Pieris rapae* PrAMPK (XP_022112548).

**Figure 3 ijms-24-08587-f003:**
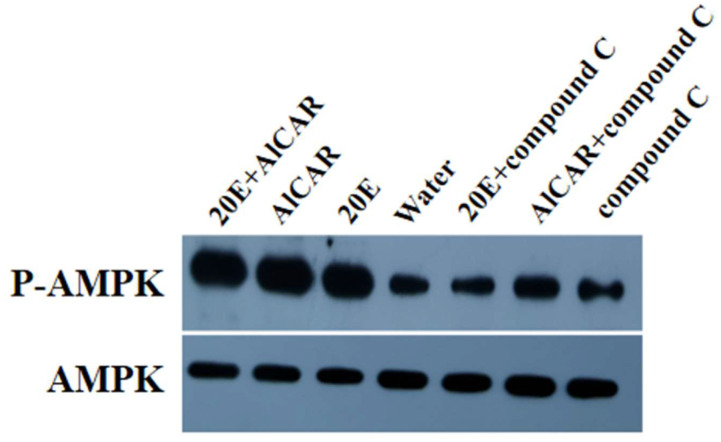
20E treatment leads to activation of AlAMPK. Fat bodies from surviving *A. lucorum* third nymphs were cultured in vitro for 1 h in the presence of 20E (1.0 μmol/L) and AlCAR (25 μmol/L), AlCAR only, 20E only, water as a control, 20E and compound C (25 μmol/L), AlCAR and compound C, and compound C only. Proteins were extracted from the fat bodies and subjected to Western blot analysis using antibodies against AlAMPK phosphorylated at threonine 172 and total AlAMPK.

**Figure 4 ijms-24-08587-f004:**
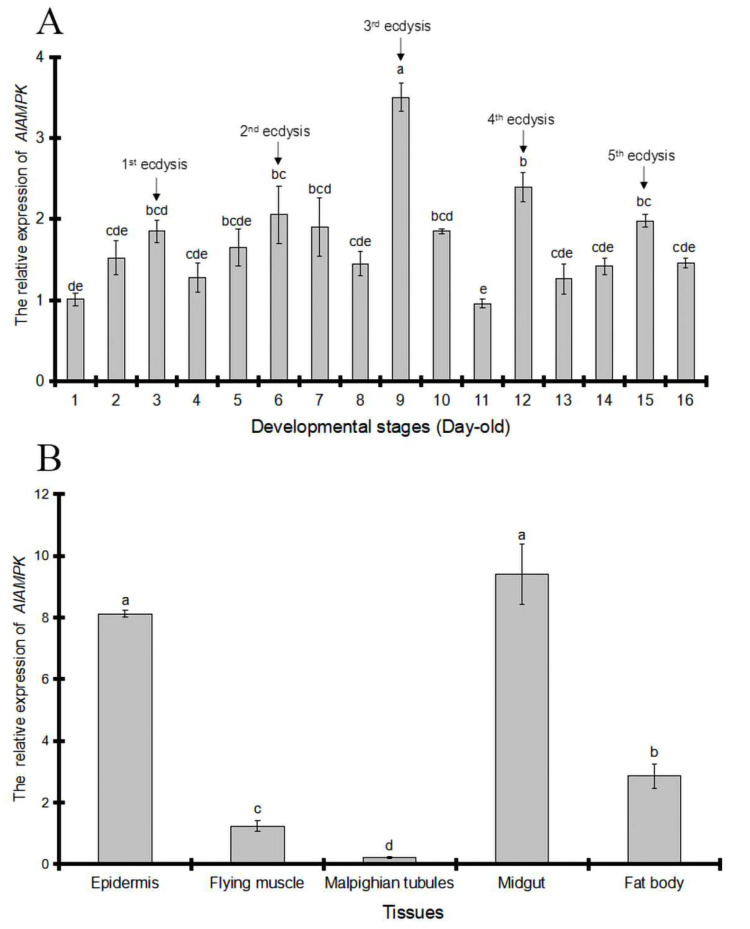
The relative developmental-stage and tissue-specific specific expression profiles of *AlAMPK* via qRT-PCR. (**A**) The relative developmental stages (day 1–day 16) expression level of *AlAMPK*. (**B**) The relative expression level of *AlAMPK* in the epidermis, flying muscle, Malpighian tubules, midgut, and fat body. The mRNA levels were normalized against the *β-actin* reference gene. Bars are mean ± SE; different letters above bars indicate highly significant differences between four treatments (*p* < 0.05, Tukey’s Honestly Significant Difference test).

**Figure 5 ijms-24-08587-f005:**
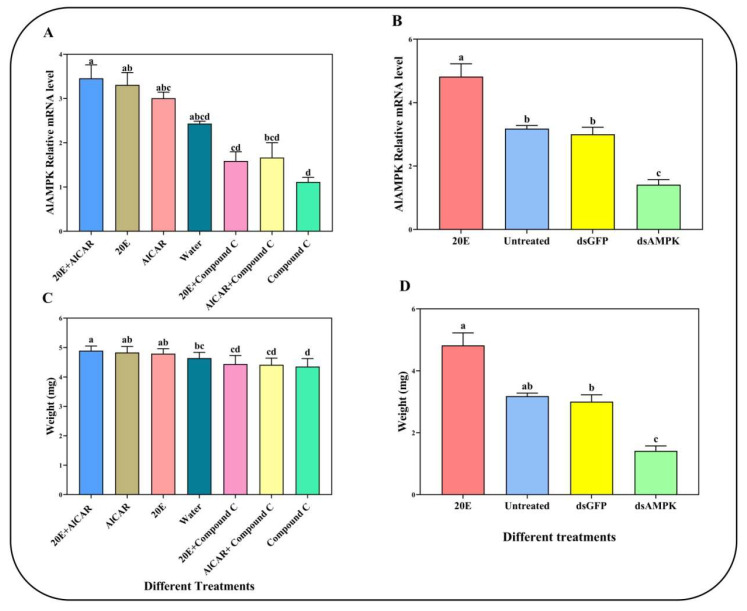
The relative mRNA expression levels of *AlAMPK* and fifth instar nymphal weight of the two different handling methods. (**A**) The relative expression level of *AlAMPK* after spraying 7 different compounds. (**B**) The relative expression level of *AlAMPK* after injecting ds*AlAMPK*, ds*GFP*, and 20E. (**C**) The fifth-instar nymphal weight after spraying 7 different compounds. (**D**) The fifth instar nymphal weight after injecting ds*AlAMPK*, ds*GFP*, and 20E. The content of 7 different compounds, dsRNA and 20E, is the same as above. The mRNA levels were normalized against the *β-actin* reference gene. Bars are mean ± SE; different letters above bars indicate highly significant differences between four treatments (*p* < 0.05, Tukey’s Honestly Significant Difference test).

**Figure 6 ijms-24-08587-f006:**
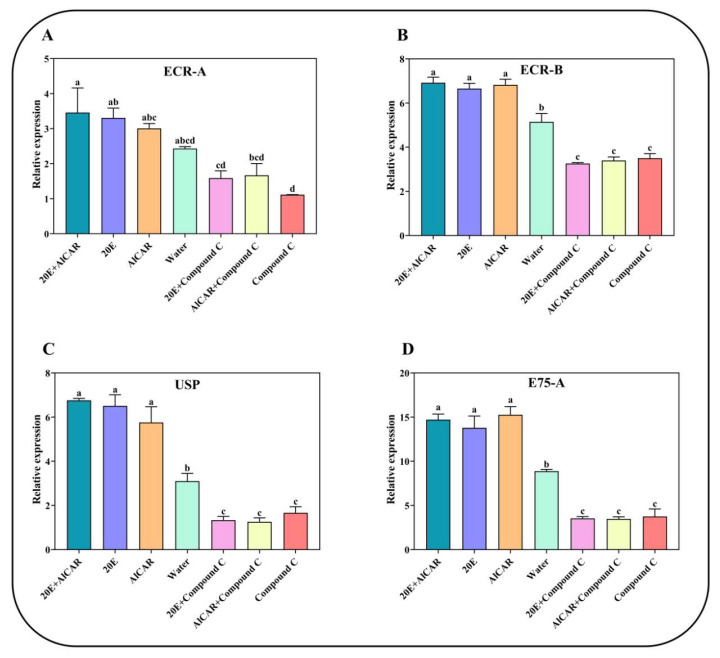
Changes in the relative expression levels of *ECR-A*, *ECR-B*, *USP* and *E75-A* (**A**–**D**) under different treatments. The mRNA levels were normalized against the β-actin reference gene. The standard errors of the means for the three biologically independent replicates are represented by error bars. Bars are mean ± SE; different letters above bars indicate highly significant differences (*p* < 0.05, Duncan’s multiple test).

**Figure 7 ijms-24-08587-f007:**
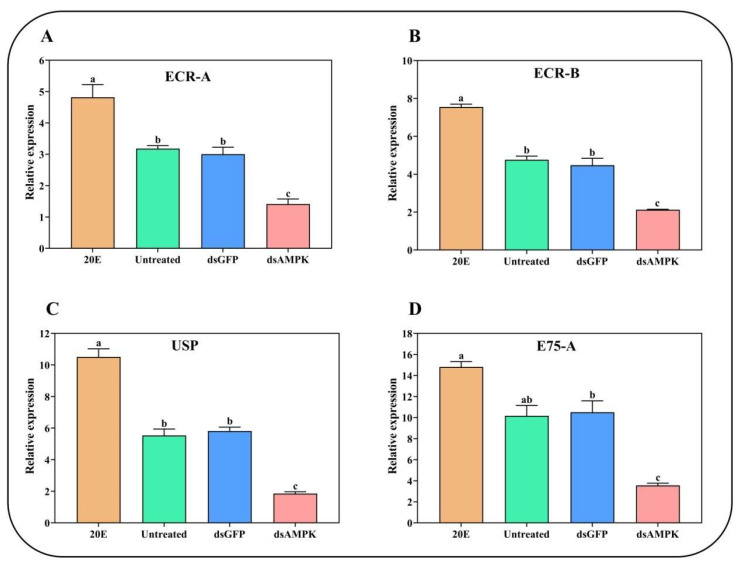
Changes in the relative expression levels of 20E-regulated genes after injecting ds*AlAMPK*, ds*GFP*, and 20E (**A**–**D**). The contents of dsRNA and 20E are the same as above. The mRNA levels were normalized against the β-actin reference gene. Bars are mean ± SE; different letters above bars indicate highly significant differences (*p* < 0.05, Duncan’s multiple test).

**Figure 8 ijms-24-08587-f008:**
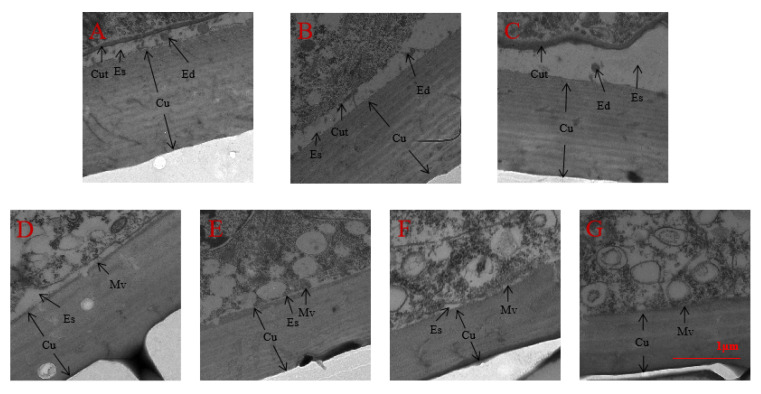
TEM observation of epidermal structure in the 3rd instar nymphs of *A. lucorum* after spraying 7 different compounds. (**A**) 20E + AlCAR (**B**) AlCAR (**C**) 20E (**D**) CK, the negative control (treatment with distilled water) (**E**) 20E + compound C (**F**) AlCAR + compound C (**G**) compound C. Cuticle: Cu, Cuticulin: Cut, Ecdysial droplet: Ed, Ecdysial space: Es, Microvilli: Mv.

**Figure 9 ijms-24-08587-f009:**
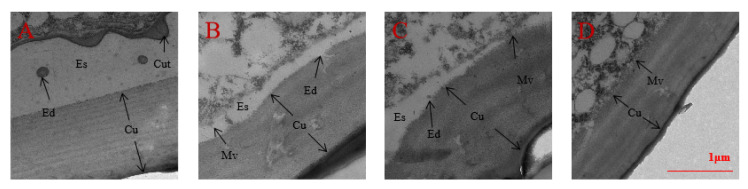
TEM observation of epidermal structure in the third instar nymphs of *A. lucorum* after dsRNA injection. (**A**) ds*AlAMPK,* (**B**) ds*GFP,* (**C**) untreated (**D**), 20E. The contents of dsRNA and 20E are the same as above. Cuticle: Cu, Cuticulin: Cut, Ecdysial droplet: Ed, Ecdysial space: Es, Microvilli: Mv.

**Table 1 ijms-24-08587-t001:** Effect of experimental treatments in *A. lucorum* nymphs on ecdysis and molting.

Experimental Treatment		1st Nymph Instar		2nd Nymph Instar		3rd Nymph Instar		4th Nymph Instar		5th Nymph Instar	
		Dt (Days)	Molting (%)	Dt (Days)	Molting (%)	Dt (Days)	Molting (%)	Dt (Days)	Molting (%)	Dt (Days)	Molting (%)
Spraying	20E + AlCAR	2.66 ± 0.12 c	98.5	2.04 ± 0.08 c	97.4	2.09 ± 0.21 c	95.2	2.22 ± 0.15 c	87.8	2.53 ± 0.26 d	84.8
	AlCAR	2.71 ± 0.14 c	99.2	2.11 ± 0.14 c	96.8	2.14 ± 0.16 c	95.8	2.19 ± 0.18 c	89.1	2.48 ± 0.14 d	86.2
	20E	2.68 ± 0.08 c	100	2.01 ± 0.12 c	97.5	2.08 ± 0.15 c	94.2	2.14 ± 0.29 c	88.7	2.43 ± 0.16 d	85.8
	Water	2.82 ± 0.13 b	99.5	2.33 ± 0.16 b	95.6	2.46 ± 0.14 b	93.2	2.35 ± 0.24 b	85.4	2.81 ± 0.23 c	82.6
	20E + compound C	2.99 ± 0.15 a	85.6	2.76 ± 0.21 a	68.5	2.98 ± 0.18 a	55.6	3.07 ± 0.29 a	52.2	3.92 ± 0.08 a	42.8
	AlCAR+ compound C	2.96 ± 0.11 a	83.2	2.81 ± 0.07 a	73.2	3.01 ± 0.18 a	50.8	2.92 ± 0.22 ab	46.2	3.59 ± 0.32 b	40.3
	compound C	3.07 ± 0.21 a	73.6	2.88 ± 0.19 a	62.8	3.42 ± 0.33 a	51.2	/	/	/	/
Injection	20E	-	-	2.12 ± 0.11 a	95.4	1.96 ± 0.15 a	93.8	2.43 ± 0.27 a	91.2	2.48 ± 0.26 a	89.6
	Untreated	-	-	2.35 ± 0.06 b	96.2	2.15 ± 0.19 b	94.2	2.47 ± 0.22 b	92.4	2.68 ± 0.19 b	88.4
	dsGFP	-	-	2.34 ± 0.08 b	94.3	2.19 ± 0.07 b	90.2	2.57 ± 0.19 b	91.8	2.62 ± 0.18 b	86.6
	dsAMPK	-	-	2.67 ± 0.28 c	62.7	3.25 ± 0.22 c	53.6	/	/	/	/

Note: Dt, Development time. Different lower letters above bars indicate highly significant differences between four treatments (*p* < 0.05, Tukey’s Honestly Significant Difference test).

## Data Availability

Not applicable.

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
