# Peer review of "Adenosine Monophosphate-Activated Protein Kinase (AMPK) Phosphorylation Is Required for 20-Hydroxyecdysone Regulates Ecdysis in Apolygus lucorum"

_ijms, 2023, doi:10.3390/ijms24108587_

Round 1

Reviewer 1 Report

The approach of the authors for the development and relevance of the subject is adequate, additionally the informations provided are orderly and coherent, i suggest a minor english editing.

For exemple, Line 52 : Please change « such as E75, E93, Br-C, and E74; finally, the nuclear receptor complex…» to « such as E75, E93, Br-C, and E74. Finally, the nuclear receptor ». 

Author Response

The approach of the authors for the development and relevance of the subject is adequate, additionally the informations provided are orderly and coherent, i suggest a minor english editing.

Response:We thank Reviewer #1 very much for the favorable and positive comments on our MS. The manuscript have edited by DBMediting and Medsic for professional English language editing services and edited by an English-speaking scientific Prof.

For exemple, Line 52 : Please change « such as E75, E93, Br-C, and E74; finally, the nuclear receptor complex…» to « such as E75, E93, Br-C, and E74. Finally, the nuclear receptor »

Response:Thank you for pointing this out.We have revised this error in revision.

Reviewer 2 Report

This is an interesting paper highlighting the role of AMPK in development and molting in A. Lucorum.

I think the work is of interest, but I found some of it in need of revision for clarity.  One key point I noticed is that the "spraying" treatment could be more clear.  The term "spraying" is used repeatedly, but in the methods section it is described as a topical application.  There are several inconsistencies that must be corrected so that the significance of the findings are clear to the reader.  I include some editorial suggestions below:

Line 11: replace “that spreads worldwide” with “with a worldwide distribution”

Line 16: add a comma after “expression”

Lines 20-21: confusing, needs revision for clarity

Lines 22-23: remove “the” before “compound C” and before “ecdysone”

Line 28: <<<”spraying” with “following spray treatment” – and are these knockdonws???

Line 30: add comma after “pathway”

Line 37: change “causing” to “causes”

Line 39: this cotton-loss clause needs revision.

Line 46: replace “tubes” with “tubules”

Line 46: change “stage” to “stages”

Line 49: italicize “in vitro”

Line 50: need space between “by binding”

Line 53: delete “which is requisite”

Line 55: change “coordinated” to “coordinates”

Line 62: change “comprising” to “comprised”

Line 76: need space between “growth and”

Line 136: change “flying” to “flight”

Line 169: change “the survived” to “surviving”

Line 244: change “generation” to “generations”

Line 271: italicize “A. lucorum”

Line 294: change “lockdown” to “knockdown”

Line 296: delete “, which”

Line 300: change “untreated” to “water control”

Line 301: change “treated” to “treatment”

Line 303: change “20 injections” to “20E injections”

Table 1 title should read: “Effect of experimental treatments in Apolygus lucorum nymphs on ecdysis and molting

Line 381- 383: not sure if you want to convey  that the 20E-induced phosphorylation effects on these processes are not well-understood, or if you want to convey that they have not been verified at all until now.

Line 444 – 448: this key findings sentence is not clear – needs revision

Author Response

This is an interesting paper highlighting the role of AMPK in development and molting in A. Lucorum.

I think the work is of interest, but I found some of it in need of revision for clarity. One key point I noticed is that the "spraying" treatment could be more clear. The term "spraying" is used repeatedly, but in the methods section it is described as a topical application. There are several inconsistencies that must be corrected so that the significance of the findings are clear to the reader. I include some editorial suggestions below:

Response:We thank Reviewer #2 very much for the favorable and positive comments on our MS. We have corrected the term “spray” in whole MS and made a revision in the methods section.

Line 11: replace “that spreads worldwide” with “with a worldwide distribution”

Response:Thank you for pointing this out. Revised.

Line 16: add a comma after “expression”

Response:Thank you for pointing this out. Revised.

Lines 20-21: confusing, needs revision for clarity

Response:Thank you for pointing this out. Revised.

Lines 22-23: remove “the” before “compound C” and before “ecdysone”

Response:Thank you for pointing this out. Revised.

Line 28: <<<”spraying” with “following spray treatment” – and are these knockdonws???

Response:Thank you for pointing this out. In this study, 20E and/or AlCAR spraying are positive influence, and Compound C is knockdowns.

Line 30: add comma after “pathway”

Response:Thank you for pointing this out. Revised.

Line 37: change “causing” to “causes”

Response:Thank you for pointing this out. Revised.

Line 39: this cotton-loss clause needs revision.

Response:Thank you for pointing this out. Revised.

Line 46: replace “tubes” with “tubules”

Response:Thank you for pointing this out. Revised.

Line 46: change “stage” to “stages”

Response:Thank you for pointing this out. Revised.

Line 49: italicize “in vitro”

Response:Thank you for pointing this out. Revised.

Line 50: need space between “by binding”

Response:Thank you for pointing this out. Revised.

Line 53: delete “which is requisite”

Response:Thank you for pointing this out. Revised.

Line 55: change “coordinated” to “coordinates”

Response:Thank you for pointing this out. Revised.

Line 62: change “comprising” to “comprised”

Response:Thank you for pointing this out. Revised.

Line 76: need space between “growth and”

Response:Thank you for pointing this out. Revised.

Line 136: change “flying” to “flight”

Response:Thank you. Flying muscle is the fixed statement in Entomology.

Line 169: change “the survived” to “surviving”

Response:Thank you for pointing this out.Revised.

Line 244: change “generation” to “generations”

Response:Thank you for pointing this out.Revised.

Line 271: italicize “A. lucorum”

Response:Thank you for pointing this out.Revised.

Line 294: change “lockdown” to “knockdown”

Response:Thank you for pointing this out.Revised.

Line 296: delete “, which”

Response:Thank you for pointing this out.Revised.

Line 300: change “untreated” to “water control”

Response:Thank you for pointing this out.Revised.

Line 301: change “treated” to “treatment”

Response:Thank you for pointing this out.Revised.

Line 303: change “20 injections” to “20E injections”

Response:Thank you for pointing this out.Revised.

Table 1 title should read: “Effect of experimental treatments in Apolygus lucorum nymphs on ecdysis and molting

Response:Thank you for pointing this out.Revised.

Line 381- 383: not sure if you want to convey that the 20E-induced phosphorylation effects on these processes are not well-understood, or if you want to convey that they have not been verified at all until now.Line 444 – 448: this key findings sentence is not clear – needs revision

Response:Thank you for pointing this out.Revised.

Reviewer 3 Report

INTRODUCTION

LINE 43 I suggest to add that ecdysone is produced through the activation of MAPK and/or PI3K pathways (https://doi.org/10.1016/j.jinsphys.2018.02.008, https://doi.org/10.1038/sdata.2016.73).

Lines 61, 68, 74, 76 Please add references to these sentences.

MATERIAL AND METHODS

Line 125 write Drosophila melanogaster.

According to the guideline reported in “Minimum Information Required for Publication of Quantitative Real-Time PCR experiments (MIQE) (Bustin et al., 2009) multiple reference genes are needed. Why did the authors use just actin? Please repeat the experiment with at least one more gene, alternatively, explain the previous validation of this gene. Moreover, the authors explain that the analysis was performed through the delta delta CT method, but, in order to use these equations, the efficiencies of the amplicons must be approximately equal and between the values of 0.8 and 1. Please explain.

FIGURE CAPTION

In the figure captions please add the statistical analysis performed.

Author Response

INTRODUCTION

LINE 43 I suggest to add that ecdysone is produced through the activation of AMPK and/or PI3K pathways (https://doi.org/10.1016/j.jinsphys.2018.02.008, https://doi.org/10.1038/sdata.2016.73).

Response:Thank you for pointing this out. We have add this information in our Revision.

Lines 61, 68, 74, 76 Please add references to these sentences.

Response:Thank you for pointing this out. Added.

MATERIAL AND METHODS

Line 125 write Drosophila melanogaster.

Response:Thank you for pointing this out. Yes, Drosophila melanogaster have already appeared in Line 64.

According to the guideline reported in “Minimum Information Required for Publication of Quantitative Real-Time PCR experiments (MIQE) (Bustin et al., 2009) multiple reference genes are needed. Why did the authors use just actin? Please repeat the experiment with at least one more gene, alternatively, explain the previous validation of this gene. Moreover, the authors explain that the analysis was performed through the delta delta CT method, but, in order to use these equations, the efficiencies of the amplicons must be approximately equal and between the values of 0.8 and 1. Please explain.

Response: We thank Reviewer #3 for the constructive comments on our manuscript. The β-actin gene is often used in the literature, the content of which in different treatments is relatively stable, and it is proved in our fluorescence quantitative experimental data, so we use β-actin as the expression level reference. Also, the β-actin has been used in our previous study (1.Tan, Y.; Xiao, L.; Sun, Y.; Zhao, J.; Bai, L. Sublethal effects of the chitin synthesis inhibitor, hexaflumuron, in the cotton mirid bug, Apolygus lucorum (Meyer-Dür).Pestic.Biochem.Phys.2014, 111, 43-50. https://doi.org/10.1016/j.pestbp.2014.04.001.  2.Tan, Y.; Xiao, L.; Sun, Y.; Zhao, J.; Bai, L.; Xiao, Y. Molecular characterization of soluble and membrane-bound trehalases in the cotton mirid bug, Apolygus lucorum.Arch. Insect Biochem. Physiol.2014b,86, 107-121. https://doi.org/10.1002/arch.21166.).

Besides, to estimate the efficiency of primers used in RT-PCR, a standard curve was constructed with 5 dilutions of cDNA (1×102, 1×101, 1×100, 1×10-1, and 1×10-2 ng) and the primer efficiency was calculated using the formula E = 10-1/SLOPE. The efficiencies ofβ-actin primers were ~ 0.86.

FIGURE CAPTION

In the figure captions please add the statistical analysis performed.

Response:Thank you for pointing this out. Added.

Reviewer 4 Report

Apolygus lucorum, plant mirid bug, is a pest that can cause major economic harm and is widespread worldwide. Adenosine monophosphate-activated protein kinase (AMPK) controls the activity of the steroid hormone 20-hydroxyecdysone (20E), which is essential for the molting and metamorphosis of insects. The authors did a great job in discovering that treatment with 20E and/or the AMPK activator AlCAR boosted the expression of AlAMPK, leading to an increase in A. lucorum's molting rate, weight, and development time. Using RNAi to inactivate AlAMPK, nymphs' weight, molt frequency, and time to develop decreased. These findings imply that by altering its phosphorylation level, AlAMPK controls hormonal signaling and regulates insect molting and metamorphosis.

Line 1-4: Please, rewrite the title. Do not start with numbers and try to make it short but informative.

Line 14: Please, replace the word “fuel” with a more appropriate word.

Line 33: Scientific name should be in italics.

Line 38: Write the BT in full form and correctly:  Bt (Bacillus thuringiensis) cotton.

Line 116: What do you mean by “Different reagents”? Make it clear.

Line 119: As you mentioned, you used DMSO to dissolve the reagents but used water as a control. In that case, how would you measure if DMSO has any effects? Why did you use water as a control?

Line 147: Did you use any other reference gene besides β-actin?

Line 159: Do you think that RNAi is the most efficient method for studying gene function? CRISPR/Cas9 genome editing tools are now widely used in studying target gene function more precisely.

Line 263: “In vitro” should be in italics.

Line 380: Write Bombyx mori in short form B. mori.

Line 408-412: Break down the sentence and write clearly.

Line 413-418: Break down the sentence and write clearly.

Line 415-418: The sentence seems you are not confident with your result. Why you used the word “speculated”? Do you not believe in the results of your experiment?

The discussion section needs to be more focused on the current study's results and the study's knowledge and implication. The conclusion needs to be drawn more carefully. Some grammatical and lexical errors need to be fixed throughout the manuscript.

Author Response

Apolygus lucorum, plant mirid bug, is a pest that can cause major economic harm and is widespread worldwide. Adenosine monophosphate-activated protein kinase (AMPK) controls the activity of the steroid hormone 20-hydroxyecdysone (20E), which is essential for the molting and metamorphosis of insects. The authors did a great job in discovering that treatment with 20E and/or the AMPK activator AlCAR boosted the expression of AlAMPK, leading to an increase in A. lucorum's molting rate, weight, and development time. Using RNAi to inactivate AlAMPK, nymphs' weight, molt frequency, and time to develop decreased. These findings imply that by altering its phosphorylation level, AlAMPK controls hormonal signaling and regulates insect molting and metamorphosis.

Response:We thank Reviewer #4 very much for the favorable and positive comments on our MS.

 Line 1-4: Please, rewrite the title. Do not start with numbers and try to make it short but informative.

Response:Thank you for pointing this out. The new title is “Adenosine Monophosphate-activated Protein Kinase (AMPK) phosphorylation is required for 20-hydroxyecdysone regulates ecdysis in Apolygus lucorum

Line 14: Please, replace the word “fuel” with a more appropriate word.

Response:Thank you for pointing this out.We think “energy” is more appropriate.

Line 33: Scientific name should be in italics.

Response:Italics.

Line 38: Write the BT in full form and correctly:  Bt (Bacillus thuringiensis) cotton.

Response:Revised.

Line 116: What do you mean by “Different reagents”? Make it clear.

Response:Revised.

Line 119: As you mentioned, you used DMSO to dissolve the reagents but used water as a control. In that case, how would you measure if DMSO has any effects? Why did you use water as a control?

Response:Thank you for pointing this out. Yes, this is a very good comments. DMSO is a conventional solvent because of its big polarity, high boiling point and stability. In the most case, DMSO was not setting for negative control, and we think water is no effect for AMPK phosphorylation. Actually, this experiment need to improved.

Line 147: Did you use any other reference gene besides β-actin?

Response:Thank you for pointing this out. We just use β-actin for the reference gene.The β-actin gene is often used in the literature, the content of which in different treatments is relatively stable, and it is proved in our fluorescence quantitative experimental data, so we use β-actin as the expression level reference. Also, the β-actin has been used in our previous study (1.Tan, Y.; Xiao, L.; Sun, Y.; Zhao, J.; Bai, L. Sublethal effects of the chitin synthesis inhibitor, hexaflumuron, in the cotton mirid bug, Apolygus lucorum (Meyer-Dür).Pestic.Biochem.Phys.2014, 111, 43-50. https://doi.org/10.1016/j.pestbp.2014.04.001.  2.Tan, Y.; Xiao, L.; Sun, Y.; Zhao, J.; Bai, L.; Xiao, Y. Molecular characterization of soluble and membrane-bound trehalases in the cotton mirid bug, Apolygus lucorum.Arch. Insect Biochem. Physiol.2014b,86, 107-121. https://doi.org/10.1002/arch.21166.).

Line 159: Do you think that RNAi is the most efficient method for studying gene function? CRISPR/Cas9 genome editing tools are now widely used in studying target gene function more precisely.

Response:Thank you for pointing this out. Yes, CRISPR/Cas9 genome editing tools are now widely used in studying target gene function, but that is huge work. In this study, RNAi was used for the gene function,and the results seem good.

Line 263: “In vitro” should be in italics.

Response:Italics.

Line 380: Write Bombyx mori in short form B. mori.

Response:Revised.

Line 408-412: Break down the sentence and write clearly.

Response:Revised.

Line 413-418: Break down the sentence and write clearly.

Response:Revised.

Line 415-418: The sentence seems you are not confident with your result. Why you used the word “speculated”? Do you not believe in the results of your experiment?

Response:Revised.

The discussion section needs to be more focused on the current study's results and the study's knowledge and implication. The conclusion needs to be drawn more carefully. Some grammatical and lexical errors need to be fixed throughout the manuscript.

Response:Revised.